# The Protective Effect of Dabigatran and Rivaroxaban on DNA Oxidative Changes in a Model of Vascular Endothelial Damage with Oxidized Cholesterol

**DOI:** 10.3390/ijms21061953

**Published:** 2020-03-13

**Authors:** Ewelina Woźniak, Marlena Broncel, Bożena Bukowska, Paulina Gorzelak-Pabiś

**Affiliations:** 1Laboratory of Tissue Immunopharmacology, Department of Internal Diseases and Clinical Pharmacology, Medical University of Lodz, Kniaziewicza 1/5, 91-347 Lodz, Poland; ewelina.wozniak@umed.lodz.pl (E.W.); marlena.broncel@umed.lodz.pl (M.B.); 2Department of Biophysics of Environmental Pollution, Faculty of Biology and Environmental Protection, University of Lodz, Pomorska 141/143, 90-236 Lodz, Poland; bozena.bukowska@biol.uni.lodz.pl

**Keywords:** dabigatran, rivaroxaban, DNA oxidative damage, vascular endothelial damage, oxidized cholesterol, reactive oxygen species, DNA bases

## Abstract

Background: Atherosclerotic plaques are unstable, and their release may result in thrombosis; therefore, currently, antiplatelet therapy with anticoagulants is recommended for the treatment of acute coronary syndrome. The aim of this study was to assess the effect of oxidized cholesterol on human umbilical vascular endothelial cells (HUVECs). The study also examines the protective and repairing effect of dabigatran and rivaroxaban in a model of vascular endothelial damage with 25-hydroxycholesterol (25-OHC). Methods: HUVECs were treated with compounds induce DNA single-strand breaks (SSBs) using the comet assay. Oxidative DNA damage was detected using endonuclease III (Nth) or human 8 oxoguanine DNA glycosylase (hOOG1). Reactive oxygen species (ROS) formation was determined using flow cytometry. Results: 25-hydroxycholesterol caused DNA SSBs, induced oxidative damage and increased ROS in the HUVECs; ROS level was lowered by dabigatran and rivaroxaban. Only dabigatran was able to completely repair the DNA SSBs induced by oxysterol. Dabigatran was able to reduce the level of oxidative damage of pyrimidines induced by oxysterol to the level of control cells. Conclusions: Observed changes strongly suggest that the tested anticoagulants induced indirect repair of DNA by inhibiting ROS production. Furthermore, dabigatran appears to have a higher antioxidant activity than rivaroxaban.

## 1. Introduction

Cardiovascular diseases, such as hypertension and atherosclerosis, are characterized by changes in the structure and activity of endothelial cells and the vasculature [1]. 25-hydroxycholesterol (25-OHC) is a type of oxidized cholesterol belonging to a subgroup of oxidized low-density lipoproteins (oxLDLs) which play major roles in atherosclerosis [2,3]. Johnson et al. [4] found 25-OHC to be upregulated in serum following the ingestion of a meal rich in oxysterols and after dietary cholesterol challenge; higher levels have also been observed in hypercholesterolemic serum than in normocholesterolemic serum [5,6]. Oxidized LDLs generate lipid-derived molecules, i.e., foam cells, that accumulate in the vascular subintimal space. The foam cells generate a fibrotic cap (atheromatous plaque) [7].

Certain common cardiovascular diseases (CVDs), such as venous thromboembolism, stroke or systemic embolism in non-valvular atrial fibrillation, can be controlled or treated using non-vitamin-K oral anticoagulants (NOAC). One such NOAC is dabigatran: a reversible competitive inhibitor of thrombin that specifically inhibits both free and clot-bound thrombin and thrombin-induced platelet activation [8]. Another is rivaroxaban: a direct factor Xa inhibitor [9].

Dabigatran and rivaroxaban are recommended for the prevention of stroke and systemic embolism in adults with non-valvular atrial fibrillation, the standard dosages are 150 mg two times per day for dabigatran and 20 mg once per day for rivaroxaban; the recommended reduced dosages are 110 mg two times per day for dabigatran and 15 mg once per day for rivaroxaban. Dose reduction recommendations for rivaroxaban are based on renal function, while dose reduction for dabigatran considers renal function, age, concomitant medications and other comorbidities [10].

Despite the current European Society of Cardiology (ESC) and Canadian guideline in patients with atrial fibrillation undergoing percutaneous coronary intervention (PCI) could not establish the noninferiority of NOAC monotherapy compared with combination therapy one year after stent implantation (The Optimizing Antithrombotic Care in Patients with Atrial Fibrillation and Coronary Stent (OAC-ALONE) trial) [11]. However, the latest research conducted by Kheiri et al. [12] showed that among patients with atrial fibrillation and stable coronary artery disease (CAD) one year beyond stent implantation, monotherapy with an NOAC was associated with similar rates of major adverse cardiovascular events (MACE) and cardioembolic events, but with a lower risk of major bleeding and hemorrhagic stroke compared to combination therapy (NOAC and antiplatelet).

Additionally, rivaroxaban and dabigatran are recommended for patients with the acute and continued treatment of deep vein thrombosis (DVT) and pulmonary embolism (PE), and for the secondary prevention of venous thromboembolisms (VTE) [10].

The Cardiovascular Outcomes for People Using Anticoagulation Strategies (COMPASS) trial showed that the combination of low-dose rivaroxaban (2 × 2.5 mg) and aspirin reduced major vascular events in patients with stable vascular disease without atrial fibrillation [13].

Interestingly, Devereaux et al. [14], in a randomized placebo-controlled trial, enrolled 1754 adult patients with myocardial injury after noncardiac surgery (MINS), and showed that dabigatran (2 × 110 mg) reduced risk for major vascular complications without increasing major bleeding.

In addition to their proven antithrombotic properties, rivaroxaban and dabigatran are known to have pleiotropic effects, one of which is reducing inflammation and oxidative activity. Song et al. [15] demonstrated that treatment with dabigatran significantly inhibited the activities of the P65 of nuclear factor κB (NF-κB), tumor necrosis factor α (TNFα), interleukin (IL)-1β and IL-6 and significantly enhanced those of the catalase and superoxide dismutase in the acute myocardial infarction in rabbits. In turn, rivaroxaban can effectively relieve the inflammatory response of endothelial cells stimulated by lipopolysaccharides (LPS), which may be related to inhibition of the NF-κB signaling pathway [16].

The present study examines the ability of anticoagulants such as dabigatran and rivaroxaban to protect against the production of reactive oxygen species (ROS) and DNA oxidative damage in a model of vascular endothelial damage with oxidized cholesterol: an area of research that is currently unknown. It employs a range of methods to provide a detailed picture of the mechanism of genotoxic action of the tested oxysterol; these methods include the use of comet assay to assess the formation of single-strand breaks (SSBs), DNA purine and pyrimidine lesions and the use of flow cytometry to detect ROS.

## 2. Results

### 2.1. Analysis of DNA Damage

#### 2.1.1. Comet Assay: Alkaline Version

Alkaline comet assay allows the detection of DNA damage manifested as SSBs, DSBs and alkali labile sites (ALSs). At a concentration of 10 µg/mL, 25-hydroxycholesterol administration increased DNA damage. This DNA damage was reduced by dabigatran and rivaroxaban; however, only 500 ng/mL dabigatran was able to completely repair the induced DNA strand breaks, i.e., return them to control levels.

Selected comets originating from the DNA of human umbilical vascular endothelial cells (HUVECs) exposed to 25-hydroxycholesterol are shown in Figure 1A,B, together with the repair of DNA damage caused by dabigatran and rivaroxaban.

#### 2.1.2. Comet Assay—Damage to Purines and Pyrimidines

25-hydroxycholesterol induced oxidative damage to purines and pyrimidines. Figure 2A shows the percentage of DNA in the comet tail derived from HUVECs exposed to 25-hydroxycholesterol; it also shows the repair of DNA damage by (Figure 2B) dabigatran and (Figure 2C) rivaroxaban following exposure to the tested agents.

### 2.2. Flow Cytometric Measurement of ROS 

#### Oxidation of H_2_DCF

The effect of 25-hydroxycholesterol (10 µg/mL) on ROS production in HUVECs was shown as changes in DCF fluorescence intensity. The addition of dabigatran and rivaroxaban reduced ROS production in cells stimulated with oxysterol (Figure 3A,B). The intensity of DCF fluorescence was assumed as 100% in negative controls (i.e., untreated HUVECs) and 350.2% in positive controls (i.e., treatment with 100 µM hydrogen peroxide). The chemicals studied were shown to generate ROS in HUVECs. 

## 3. Discussion

The present study is the first to investigate the effect of oxidized cholesterol 25-hydroxycholesterol, on endothelial cells, and two selected anticoagulants: dabigatran and rivaroxaban. The concentrations of the two anticoagulants are comparable to therapeutic variable: 100 ng/mL and 500 ng/mL (229 nM, 1147 nM) for dabigatran, and 100 ng/mL and 500 ng/mL (159 nM, 796 nM) for rivaroxaban. The plasma concentration of rivaroxaban is reported to peak about 141–318 ng/mL (224–506 nM) following a dose of 20 mg [17]. In addition, dabigatran plasma concentrations about 200 ng/mL (458 nM) were observed in patients with AF treated for prevention of stroke and systemic embolism with 150 mg dabigatran etexilate twice daily [18]. Hawes et al. [19] determined the performance of various coagulation assays in patients treated with dabigatran 150 mg twice daily. ‘Therapeutic range’ was defined as the range of plasma dabigatran concentrations determined by mass spectrometry between the 2.5th and 97.5th percentiles of all values. The therapeutic range was 27–411 ng/mL.

Our results indicate that 25-hydroxycholesterol did not significantly affect the viability or apoptosis of endothelial cells. Similarly, no changes in viability or apoptosis were observed among endothelial cells following treatment with a mixture of palmitic acid (800 µM) and 25-hydroxycholesterol (10 µg/mL) [20]. Similar results were obtained in primary human aortic endothelial cells (HAECs) incubated with 25-hydroxycholesterol (10 µg/mL) [21]. Moreover, 25-OHC in a range of 0–10 µM had no effects on proliferation or apoptosis of gastric cancer cell lines (AGS and MGC-803) in vitro [7]. Another study on purified Leydig cells of mature rats found apoptotic effects to be induced by 24-h incubation with 25-OHC, but only at a high concentration (125 µM) [22].

Literature data suggest that neither dabigatran and rivaroxaban increase apoptosis in endothelial cells; on the contrary, dabigatran has been found to significantly reduce apoptosis in acute myocardial infarction (AMI) vehicle rabbits [15]. Rivaroxaban significantly reduced the number of apoptotic cells in human umbilical vein endothelial cells (HUVECs) treated with LPS [16]. Similar results were obtained by Álvarez et al. [23], who report that rivaroxaban (1 nM–1 μM) enhanced viability and showed a significant and dose-dependent positive effect on HUVEC growth that was inhibited by BC-11-hydroxibromide, an inhibitor of u-PA. However, our present findings indicate that these anticoagulants do not appear to demonstrate any effect on endothelial cell viability or apoptosis at the concentrations used in these experiments.

Literature data suggest that oxysterols are associated with various cardiovascular, metabolic, neurodegenerative and cancerous pathologies. A strong correlation has been observed between cholesterol 25-hydroxylase, i.e., *CH25H*, mRNA expression and that of fibrosis markers in human intestinal samples from Crohn’s disease patients [24]. In addition, the oxysterols 25-hydroxycholesterol and 7-ketocholesterol induce damage to the barrier tissues such as the vascular endothelium and intestinal epithelium, which may lead to disturbances of local immune homeostasis [21].

As 25-hydroxylolesterol may have an adverse effect on various tissues, the present study also assesses the genotoxicity of this compound. Our findings indicate that 25-hydroxycholesterol induced DNA SSBs, and that this effect was reversed by treatment with dabigatran or rivaroxaban. It is also worth noticing that only 500 ng/mL dabigatran was able to fully repair the DNA strand breaks induced by oxysterol, indicated by a return to control values.

Single-strand breaks can also be induced by oxidative DNA damage to purines and pyrimidines. Our results indicate that 25-hydroxycholesterol induced oxidative damage to nitrogenous bases, and that that vascular endothelial cells showed a greater tendency to suffer oxidative damage to purines than pyrimidines following oxysterol stimulation, but this difference was not statistically significant. The most commonly observed DNA lesion induced by oxidative stress is 8-oxo-7,8-dihydroguanine (8-oxodG). 8-oxodG has a high potential for mutation, realized by the misincorporation of an adenine instead of cytosine, i.e., a G:C→T:A transversion mutation [25].

Oxidative DNA damage is often caused by the presence of oxidative stress, associated with increased production of reactive oxygen species (ROS) [26]. Oxidative stress may be elevated in atheroma plaques due to the presence of increased levels of 7b-HC and 7-KC, which stimulate ROS production and deplete cellular antioxidant reserves [27].

In turn, among the endothelial cells prestimulated with oxysterol, those that were incubated with dabigatran and rivaroxaban, at all tested concentrations, displayed less damage to both purines and pyrimidines. It is also important to note that only 100 and 500 ng/mL dabigatran was able to reduce the level of oxidative damage to pyrimidines induced by oxysterol to control levels. This reduction in oxidative DNA damage may be associated with the strong antioxidant activity of the anticoagulants.

Previously, dabigatran was found to significantly enhance catalase and superoxide dismutase activity and suppress inducible nitric oxide synthase (iNOS) in a rabbit model of acute myocardial infarction (AMI); the AMI was induced by an intravenous bolus (0.5 mg/kg) and concomitant infusion (0.15 mg/kg/h) [15]. In addition, dabigatran treatment (1.2 g/kg/d) significantly reduced free radical excess in the vessel wall, improved endothelial function and decreased atherosclerosis in ApoE−/− mice via thrombin inhibition. Similarly, dabigatran (1 nM) induced thrombin inhibition has also been found to block the increase in inflammatory protein expression and ROS generation evoked by hypoxia in endothelial cell culture [28]. Administration of dabigatran to transgenic Alzheimer’s disease mice diminishes ROS levels in the brain and reduces cerebrovascular expression of inflammatory proteins [29].

It has been proposed that rivaroxaban treatment may indirectly reduce ROS generation. Rivaroxaban treatment (300 nM) has been found to suppress monocyte chemoattractant protein 1 gene (MCP-1) mRNA levels in tubular cells via inhibition of factor Xa (FXa), and MCP-1 is believed to potentiate FXa-driven ROS generation [30]. Rivaroxaban has also been found to suppress FXa-driven oxidative stress in human proximal tubular cells (HK-2 cells). FXa is also believed to activate PAR2 and cause inflammation, tissue fibrosis and cell proliferation and activate platelets via interaction with glycoprotein (GPVI) [31]. Cammisotto et al. [32] report that GPVI activation is associated with a burst of Nox2-derived oxidative stress, which has been implicated in PA and eicosanoid formation. Rivaroxaban (30–60 ng/mL) has been found to inhibit the activation of human platelets stimulated with convulxin (0.1 μg/mL) by lowering GPVI activation and reducing oxidative stress.

Under physiological conditions, the differences in efficiency of both anticoagulants would be dependent on the physiological response of endothelial cells to thrombin and Xa factor. Despite some molecular similarities between the thrombin and the Xa coagulation factor enzymes, these serine proteinases have a diverse place in the physiology of the blood coagulation cascade as well as in the vascular physiology [33,34,35]. Thrombin is an effector enzyme, while the coagulation factor X is an intermediate in the cascade and its activation is dependent not only on the tenase complex, but also on thrombin action (through feedback). Although both the thrombin and the Xa factor share part of biochemical pathways, their impact of cellular response may be different. However, in an experimental model of endothelial cells culture (without the stimulation by thrombin), these differences may be rather dependent on the individual influence of these anticoagulants, and not only on the inflammatory response of HUVECs, but also on oxidative stress, which is associated with the inflammation. Molecular mechanisms of these actions are only partly recognized. For example, in vitro, rivaroxaban was able to suppress the expression of proinflammatory markers in endothelial cells to a similar extent to dabigatran [36].

Furthermore, the major pathway tasked with the removal of oxidative DNA damage, and hence maintaining genomic integrity, is base excision repair (BER) with the involvement of DNA glycosylases in the first step and other enzymes in subsequent steps [37]. Perhaps, the tested anticoagulants affect the BER repair mechanisms in some way by increasing the activity of the enzymes involved in it. According to the literature, both rivaroxaban and dabigatran displayed some antioxidant effects, but no detailed information on their efficiency has been found so far. Thus, this issue requires further studies.

This is, to our knowledge, the first study trying to assess the protective and repairing effect of anticoagulants such as dabigatran and rivaroxaban on the production of reactive oxygen species and DNA oxidative damage in a model of vascular endothelial damage with oxidized cholesterol. The limitation of our work is to use only vascular endothelial cells. Generally, in vitro studies are based on single-cell cultures, which exclude interactions between different cell types. As demonstrated in the literature, the interaction between endothelial cells and smooth muscle cells (SMCs) is an essential maturation process in physiological conditions [38,39]. Proliferation and migration of smooth muscle cells occur during the early stage of atherosclerotic [40]. Numerous cell types are involved in atherogenesis and a complex network of transcription factors and proteins is involved in this process [41]. Hence, as the key parts of the vascular wall, the interaction between endothelial cells and smooth muscle cells with oxidized cholesterol needs to be further studied. An important aspect of future research would also be the verification of the protective and repairing effect of anticoagulants in model endothelial cells in co-culture with vascular smooth muscle cells.

The main limitation of our work is a measurement of only total cellular ROS without mitochondrial ROS. Mitochondria are the major source of intracellular ROS [42,43]. Xie et al. [44] suggested that reducing mitochondrial ROS production attenuates atrial diastolic SR Ca^2+^ leak and prevents AF. Pharmacological targeting of genetically inhibiting mitochondrial ROS production prevents AF providing mechanistic insights that could lead to new therapeutic targets for AF. Considering that dabigatran is also used in patients with AF treated for the prevention of stroke and systemic embolism, it would be important to verify mitochondrial oxidative stress and possible protective effects of dabigatran and rivaroxaban on this process.

Another limitation of our work is the lack of measurement of 8-oxoguanine (8-oxoG), of which accumulation in mitochondrial DNA causes mitochondrial dysfunction. 8-Oxoguanine glycosylase (OGG1) plays a protective role in atherogenesis by preventing excessive inflammasome activation [45]. In addition, Shah at al. [46] suggest that reducing oxidative damage by rescuing OGG1 activity reduces plaque development, indicating the detrimental effects of 8-oxoG on VSMC function. Therefore, 8-oxoG should be determined in both endothelial cells and smooth muscle cells with oxidized cholesterol and the effect of anticoagulants in this model must be checked.

In summary, our present findings also indicate that dabigatran and rivaroxaban treatment was associated with lower ROS production in the tested oxysterol-stimulated vascular endothelial cells. This effect was not dose-dependent: both tested concentrations of dabigatran and rivaroxaban reduced ROS production. Our results suggest that 25-OHC can induce DNA damage indirectly through ROS-mediated effects. In addition, such DNA damage may be induced indirectly by the formation of by-products with oxidative damaging potential. It has been widely accepted that ROS may cause DNA damage [47]. The tested anticoagulants dabigatran and rivaroxaban were found to counteract this DNA damage by inhibiting ROS production.

## 4. Materials and Methods

### 4.1. Chemicals

Human umbilical vascular endothelial cells (HUVECs), trypsin with EDTA, trypsin neutralizing solution, endothelial cell growth medium-2 (EGM-2) with hydrocortisone, hFGF-B, vascular endothelial growth factor (VEGF), R3-IGF-1, ascorbic acid, hEGF, GA-1000, heparin and fetal bovine serum (FBS), were purchased in Lonza (Basel, Switzerland). Endonuclease III (Nth) and human 8-oxoguanine DNA glycosylase (hOGG1) were acquired in New England Biolabs (Ipswich, MA, USA). 2′,7′-dichlorodihydrofluorescein diacetate (H_2_DCFDA) was purchased from Invitrogen (Waltham, Massachusetts, USA). Low melting point (LMP), normal melting point (NMP) agarose, 4′,6-diamidino-2-phenylindole (DAPI), dabigatran, rivaroxaban and 25-hydroxycholesterol (25-OHC) were bought from Sigma-Aldrich (St. Louis, MO, USA) while other chemicals were purchased from Carl Roth GmbH + Co. KG (Karlsruhe, Germany) and POCh (Gliwice, Poland) and were of analytical grade.

### 4.2. Cells

Human umbilical vascular endothelial cells (HUVECs) were cultured in endothelial cell growth medium-2 (EGM-2) supplemented with 10% fetal bovine serum (FBS), hydrocortisone, hFGF-B, vascular endothelial growth factor (VEGF), R3-IGF-1, ascorbic acid, hEGF, GA-1000, heparin, penicillin (100 U/mL) and streptomycin (100 µg/mL) at 37 °C, 5% CO_2_. After reaching 80–90% confluence, the HUVECs were trypsinized with 0.05% trypsin with 0.02% EDTA for 3 min and then neutralized by trypsin neutralizing solution for further experiments.

The viability of the cells was over 98% (Table 1). The Trypan Blue dye exclusion test was used to determine the number of viable cells. Data are expressed as mean ± SD.

The investigations were approved by the Bioethics Committee of the Medical University of Lodz No. RNN/363/19/KE).

### 4.3. Cell Treatment

Both trypsinized HUVECs were separately seeded on 24-well plates at a density of 100,000 cells per well in 600 µL proper medium. After reaching 80–90% confluence, HUVECs were stimulated with 25-hydroxycholesterol (10 µg/mL) for four hours. After incubation, the cells were centrifuged, the compound was discarded, and HUVECs were stimulated with dabigatran and rivaroxaban (100 and 500 ng/mL) for 24 h. The cells were incubated with compounds for one hour to determine reactive oxygen species content. After incubation, the cells were centrifuged, the compounds were discarded, and the cells were resuspended in EGM-2 medium.

### 4.4. Comet Assay

Damage to DNA provoked by 25-hydroxycholesterol was assessed by means of single-cell gel electrophoresis (comet assay). In this technique, the cells are immersed in low melting point agarose (LMP), placed on microscopic slides, and then lysed. As a result, the released DNA is submitted to electrophoresis in alkaline conditions (pH > 13). The comet assay enables the identification of SSBs and DSBs, as well as alkali labile sites (ALSs). Modification of the comet assay with repair enzymes enables examination of the purine and pyrimidine lesions.

#### 4.4.1. Alkaline Version

##### Slide Preparation and Lysis

The comet assay was performed under alkaline conditions according to Singh et al. [48] with some modifications [49] as described previously by Blasiak and Kowalik [50]. A freshly prepared cell suspension in 0.75% LMP agarose dissolved in PBS was layered onto microscope slides (Superior, Germany), which were pre-coated with 0.5% NMP agarose. Then, the cells were lysed for one hour at 4 °C in a buffer containing 2.5 M NaCl, 0.1 M EDTA, 10 mM Tris, 1% Triton X-100 and pH 10. After cell lysis, the slides were placed in an electrophoresis unit. The DNA was allowed to unwind for 20 min in the solution containing 300 mM NaOH and 1 mM EDTA, pH > 13. Each experiment concerning DNA damage included a positive control. Hydrogen peroxide at 20 µM was selected to induce DNA SSBs (the cells were incubated with H_2_O_2_ for 15 min on ice).

##### Electrophoretic separation and staining

Electrophoretic separation was performed in the solution containing 30 mM NaOH and 1 mM EDTA, pH > 13 at an ambient temperature of 4 °C (the temperature of the running buffer did not exceed 12 °C) for 20 min at an electric field strength of 0.73 V/cm (28 mA). Then, the slides were washed in water, drained, stained with 2 µg/mL DAPI and covered with cover slips. In order to prevent additional DNA damage, the procedure described above was conducted under limited light or in the dark.

##### Comet analysis

The comets were observed at 200× magnification in a fluorescence microscope (Zeiss Axio Scope.A1) connected with an Axiocam 305 color video camera (Carl Zeiss AG, Oberkochen, Germany). The microscope was connected to a desktop PC equipped with Lucia-Comet v. 7.60 software (Laboratory Imaging, Praha, Czech Republic).

Fifty images (comets) were randomly selected from each sample and the mean value of DNA in the comet tail was taken as an index of DNA damage (expressed in percent). For one experiment, two parallel tests with aliquots of the cell sample were performed for a total number of 100 comets. A total number of 400 comets was recorded to calculate the mean ± SEM.

#### 4.4.2. DNA Repair Enzyme Treatment: Purine and Pyrimidine Detection

Detection of oxidative DNA damage was conducted using endonuclease III (Nth) and human 8-oxoguanine DNA glycosylase (hOGG1). Nth and hOGG1 are capable of converting oxidized pyrimidines and purines, respectively, into DNA single-strand breaks (SSBs) that can be determined by the comet assay. After lysis, the slides were subjected to three five-minute washes using an enzyme buffer (40 mM HEPES-KOH, 0.5 mM Na_2_EDTA, 0.1 M KCl, 0.2 mg/mL BSA; pH 8). Next, agarose on slides was covered with a volume of 50 µL of buffer consisting of 1 U of Nth or hOGG1 or without the enzyme. Then, the slides were covered with cover glasses and were incubated for 30 min at 37°C in a moist chamber. In the next step, the cover glasses were removed and the slides were placed in an electrophoresis unit [51]. DNA was allowed to unwind for 20 min in a solution consisting of 300 mM NaOH and 1 mM EDTA (pH > 13).

The degree of purine and pyrimidine oxidation in the test cultures was compared with that of the positive controls, i.e., the cell cultures incubated with 20 µM hydrogen peroxide for 15 min on ice before enzyme treatment.

### 4.5. Oxidation of H_2_DCF

In order to measure the production of reactive oxygen species (ROS), the fluorescence of the probe 6-carboxy-2′,7′-dichlorodihydrofluorescein diacetate (H_2_DCFDA) was measured. The increase in fluorescence of DCF (a marker of probe oxidation) was measured by a flow cytometer (LSR II. Becton Dickinson) at excitation/emission wavelengths at 488 nm and 530 nm, respectively. Positive controls were created by adding 100 µM hydrogen peroxide to the cell suspension for 10 min on ice. The HUVECs were treated with DCF at a final concentration of 2 µM and incubated for 15 min at 37 °C in total darkness. The data was recorded for a total of 10,000 events per sample. Data are expressed as mean ± SD.

### 4.6. Statistical Analysis

All analyses were conducted with STATISTICA v 12.5 software (2000 Stat-Soft, Inc., Tulsa, OK, USA). The normality of distribution was determined using the Shapiro-Wilk test, and the homogeneity of variance by the Brown–Fisher test. Tukey’s test was used as a post-hoc test. The sample size and the statistical power were checked for all data. Differences were regarded as significant for *p* < 0.05. The individual analysis was performed in four independent experiments, while each experiment was repeated twice or three times depending on the method.

## 5. Conclusions

25-hydroxycholesterol induced DNA SSBs, as well as oxidative damage to purines and pyrimidines and increased ROS in human HUVECs. All damage was ameliorated by dabigatran or rivaroxaban treatment. The observed changes strongly suggest that the two anticoagulants indirectly support DNA repair by inhibiting ROS production; in addition, dabigatran appears to demonstrate greater antioxidant activity than rivaroxaban.

## Figures and Tables

**Figure 1 ijms-21-01953-f001:**
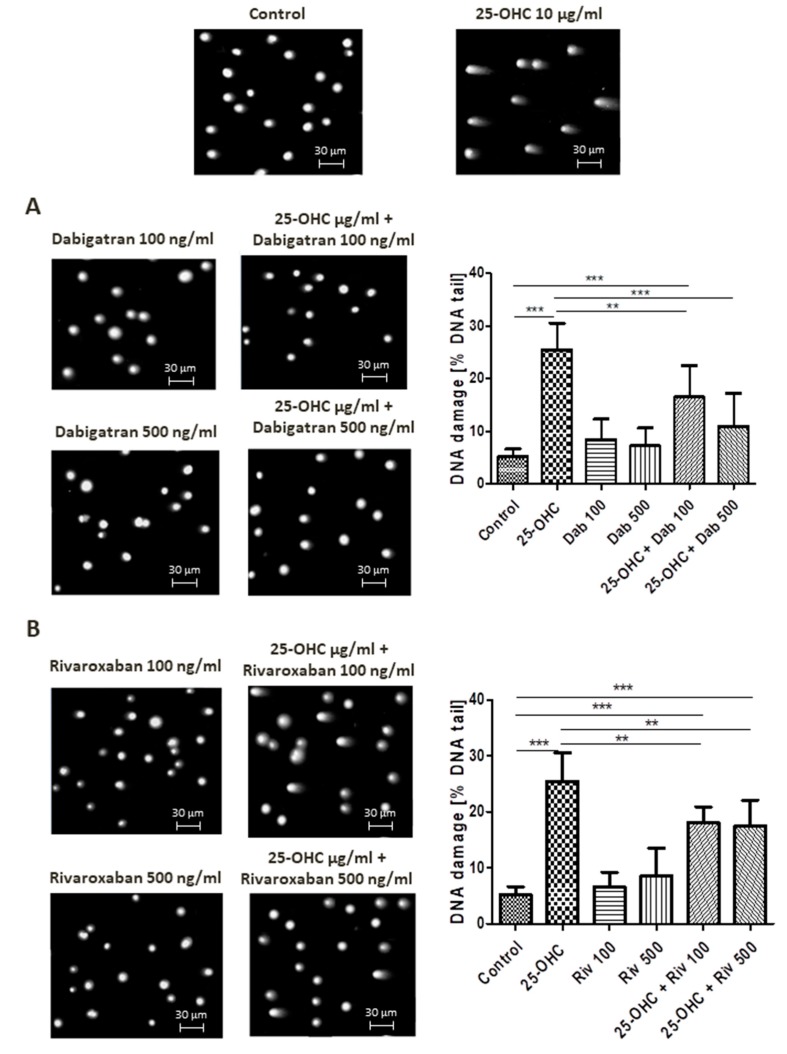
Selected photographs of DNA (comets) of human umbilical vascular endothelial cells (HUVECs) incubated with 25-hydroxycholesterol (10 µg/mL), (**A**) dabigatran and (**B**) rivaroxaban (100 ng/mL and 500 ng/mL). The photos were achieved using a fluorescent microscope with 200× magnification. Scale bars in images were added. The level of DNA strand breaks in HUVECs was determined by single-cell gel electrophoresis (comet assay). DNA damage in HUVECs was induced by 25-hydroxycholesterol, the repair of DNA damage in HUVECs was induced by (**A**) dabigatran and (**B**) rivaroxaban. DNA damage was measured as a percentage of DNA in the comet tails. Each experiment included a positive control (PC) treated with hydrogen peroxide at 20 μM (% DNA tail: 45.8 ± 8.5). The number of cells scored for each slide was 100. Mean ± SEM was calculated from four individual experiments (400 comets). Significant differences from negative control are indicted by ** *p* < 0.01; *** *p* < 0.001. Statistical analysis was conducted using one-way ANOVA and a post-hoc Tukey’s test.

**Figure 2 ijms-21-01953-f002:**
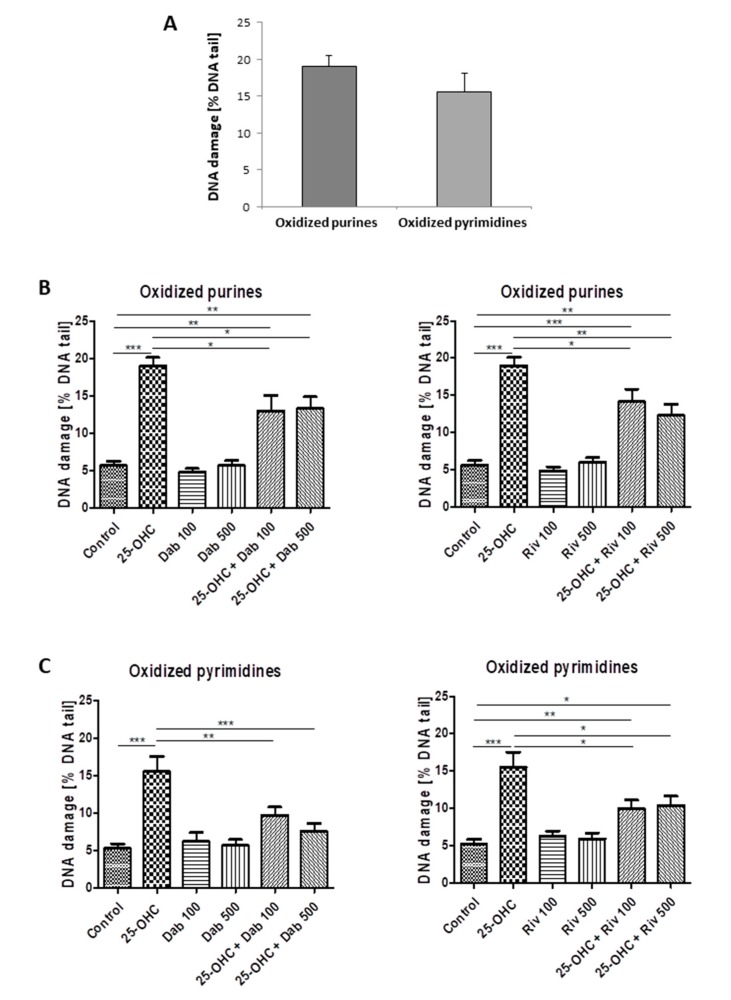
The level of DNA pyrimidine and purine oxidation in HUVECs (analysis by alkaline comet assay using endonuclease III (Nth)) or human 8 oxoguanine DNA glycosylase (hOOG1) was induced by 25-hydroxycholesterol (10 µg/mL) (**A**). The repair of DNA damage in cells was induced by (**B**) dabigatran (100 ng/mL and 500 ng/mL) and (**C**) rivaroxaban (100 ng/mL and 500 ng/mL). Each experiment included a positive control (PC) which concerned the cells incubated with hydrogen peroxide at 20 µM for 15 min on ice and subsequently treated with the enzymes. The value of DNA in the figure captions for comet tail in the presence of either enzyme for all groups was reduced by the value obtained in the comet assay without any enzyme and the value for enzyme buffer only. The number of cells scored from each slide was 100. The mean value for 100 cells analyzed in each treatment in four independent experiments (400 total cells) was recorded. Mean ± SEM was calculated from four individual experiments (400 comets) and was significantly different from negative controls at * *p* < 0.05; ** *p* < 0.01; *** *p* < 0.001. Statistical analysis was conducted using one-way ANOVA and post-hoc Tukey’s test.

**Figure 3 ijms-21-01953-f003:**
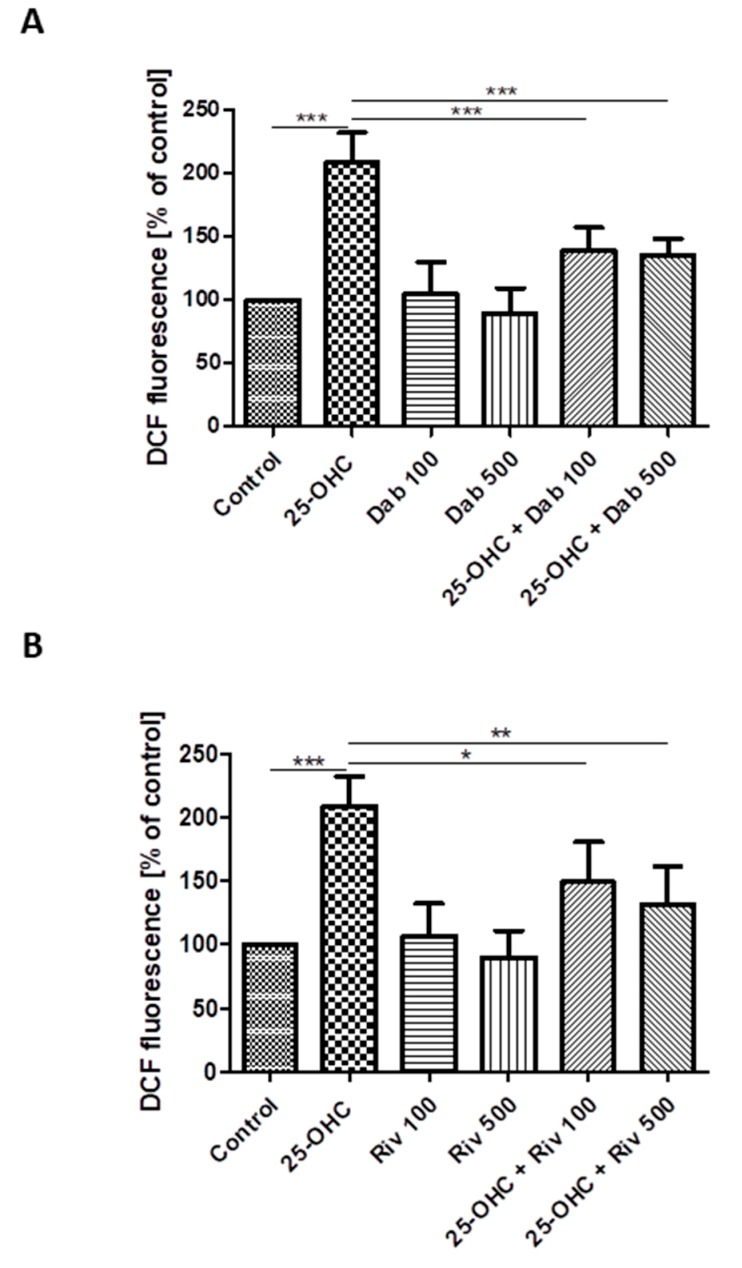
Changes in reactive oxygen species (DCF fluorescence) in HUVECs incubated with 25-hydroxycholesterol (10 µg/mL), (**A**) dabigatran (100 ng/mL and 500 ng/mL) and (**B**) rivaroxaban (100 ng/mL and 500 ng/mL). Mean ± SD calculated from nine individual experiments. Significant differences from negative controls are indicated by * *p* < 0.05; ** *p* < 0.01; *** *p* < 0.001. Statistical analysis was conducted using one-way ANOVA and post-hoc Tukey’s test.

**Table 1 ijms-21-01953-t001:** The level in the viability level of human HUVECs was determined by the Trypan Blue dye exclusion test. HUVECs were induced by 25-hydroxycholesterol (10 µg/mL), dabigatran (100 ng/mL and 500 ng/mL) and rivaroxaban (100 ng/mL and 500 ng/mL). Mean ± SD was calculated from nine individual experiments. Significant difference from negative controls was indicated by * *p* < 0.05. Statistical analysis was conducted using one-way ANOVA and post-hoc Tukey’s test.

Compounds	Concentration	Cell Viability [%]	ANOVA I
Control	0 µg/mL	98.1 ± 1.1	-
25-hydroxycholesterol	10 µg/mL	97.8 ± 1.8	*p* > 0.05
Dabigatran	100 ng/mL	98.7 ±1.3	*p* > 0.05
	500 ng/mL	98.5 ± 0.5	
Rivaroxaban	100 ng/mL	99.1 ± 1.6	*p* > 0.05
	500 ng/mL	98.9 ± 1.0	

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
