# Peer review of "The Protective Effect of Dabigatran and Rivaroxaban on DNA Oxidative Changes in a Model of Vascular Endothelial Damage with Oxidized Cholesterol"

_ijms, 2020, doi:10.3390/ijms21061953_

Round 1
Reviewer 1 Report
In the introduction the explanation of rivaroxaban use as anti-platelet therapy is incorrect. Rivaroxaban only inhibits factor Xa as the authors state and thus it is an anticoagulant. It is was tested as an add-on to antiplatelet therapy (in this case aspirin) but it does not inhibit platelets and is not an anti-platelet agent. This introductory paragraph should be rephrased.
The controls in figure 2A, show that oxidation of purine and pyrimidine control in the absence of 25-OHC. However, in figure 2B and 2C, the control bar for 25-OCH seems to be identical for both purine and pyrimidine and different from the values in 2A, where pyrimidines are somewhat less than purines. The authors should please explain.
First paragraph of the discussion, the levels of anticoagulant should be better referenced. For instance for the levels quoted from the 10 mg dose of rivaroxaban, this is not the dose used for stroke prevention in atrial fibrillation. Here the approved dose is 20 mg. Dabigatran levels have no citation. Consider Hawes et al J Thromb Hemost 11:1493, 2013 for dabigatran.
Do the authors have any theories why direct thrombin inhibition would completely restore DNA damage, while direct factor Xa inhibition only partially did this?
Manuscript should be rechecked by an English native speaker for grammar.
Reviewer 2 Report
The Authors should provide (or at least discuss in the limitations) a measurement of mitochondrial ROS (e.g. MitoSOX: Marks et al. Sci Rep. 2015 Jul 14;5:11427, PNAS. 2015 Sep 8;112:11389-94) not only global cellular ROS. Moreover, oxidative DNA damage (8-OxoG) should be assessed as well (Nakabeppu Y et al. Sci Rep 2016;6:22086).
The importance of rivaroxaban (Ann Intern Med. 2020 Jan 21;172(2):JC6. doi: 10.7326/ACPJ202001210-006) and dabigatran (Ann Intern Med. 2018 Oct 16;169(8):JC41. doi: 10.7326/ACPJC-2018-169-8-041) in cardiovascular medicine should be better addressed.
A proper dimensional bar should be provided for each picture.
The strengths and limitations of the study should be deeply addressed, taking into account sources of potential bias or imprecision: Discuss both direction and magnitude of any potential bias.
Round 2
Reviewer 2 Report
The Authors were only partially responsive.
For instance, a proper dimensional bar has not been provided.
Round 3
Reviewer 2 Report
-